# Monocyte and Macrophage in Neuroblastoma: Blocking Their Pro-Tumoral Functions and Strengthening Their Crosstalk with Natural Killer Cells

**DOI:** 10.3390/cells12060885

**Published:** 2023-03-13

**Authors:** Chiara Vitale, Cristina Bottino, Roberta Castriconi

**Affiliations:** 1Department of Experimental Medicine (DIMES), University of Genoa, 16132 Genoa, Italy; 2Laboratory of Clinical and Experimental Immunology, IRCCS Istituto Giannina Gaslini, 16147 Genova, Italy

**Keywords:** macrophages, natural killer cells, neuroblastoma, IL-18

## Abstract

Over the past decade, immunotherapy has represented an enormous step forward in the fight against cancer. Immunotherapeutic approaches have increasingly become a fundamental part of the combined therapies currently adopted in the treatment of patients with high-risk (HR) neuroblastoma (NB). An increasing number of studies focus on the understanding of the immune landscape in NB and, since this tumor expresses low or null levels of MHC class I, on the development of new strategies aimed at enhancing innate immunity, especially Natural Killer (NK) cells and macrophages. There is growing evidence that, within the NB tumor microenvironment (TME), tumor-associated macrophages (TAMs), which mainly present an M2-like phenotype, have a crucial role in mediating NB development and immune evasion, and they have been correlated to poor clinical outcomes. Importantly, TAM can also impair the antibody-dependent cellular cytotoxicity (ADCC) mediated by NK cells upon the administration of anti-GD2 monoclonal antibodies (mAbs), the current standard immunotherapy for HR-NB patients. This review deals with the main mechanisms regulating the crosstalk among NB cells and TAMs or other cellular components of the TME, which support tumor development and induce drug resistance. Furthermore, we will address the most recent strategies aimed at limiting the number of pro-tumoral macrophages within the TME, reprogramming the TAMs functional state, thus enhancing NK cell functions. We also prospectively discuss new or unexplored aspects of human macrophage heterogeneity.

## 1. Introduction

In the past decade, we have witnessed a scientific breakthrough with immunotherapy that has grabbed the cover of the most prestigious scientific journals due to its significant impact on patients’ survival [1,2]. In particular, the blockade of PD-1/PD-Ls immune-checkpoint molecules represents the gold standard immunotherapeutic approach in different cancers including melanoma [3,4] and non-small cell lung cell cancer (NSCLC). Currently, it has become the first-line monotherapeutic approach in some types of cancer such as advanced NSCLC lacking mutations in targetable Tyrosine Kinases [5,6]. However, despite the impressive results seen in a percentage of patients, others showed unresponsiveness or resistance to this immunotherapy. This represents a significant challenge for further application and forces the exploration of new combined strategies to overcome the failures [7]. Moreover, some patients, despite having clinical benefits, developed autoimmune diseases that forced the interruption of therapy with fatal tumor recurrence. To understand what mostly impacts the therapy effectiveness and disease’s course, different aspects of the tumor landscape have been analyzed, including patients’ microbiota [8,9,10], and the grade and quality of tumor infiltration by different immune cell types, termed “Immunoscore” [11,12,13,14]; these include macrophages and natural killer (NK) cells showing prognostic and predictive significance also in terms of immunotherapies’ efficacy [15,16].

Macrophages are the most abundant immune cells in the tumor microenvironment (TME). They can originate from resident macrophages present in the early phases of tumor development, and from recruited monocytes undergoing differentiation-polarization under the influence of the signals occurring in the TME. Both in mice and human, M1 and M2 indicate macrophages lying at opposite ends of the polarization process, endowed with proinflammatory anti-tumor or reparative pro-tumoral functions, respectively. However, this seminal dichotomy has been overcome by a more complex description of macrophage polarization obtained by in vitro studies. M2a, b, c, and d populations can be further distinguished based on the expression of surface and secreted molecules. M2a, derived from IL-4 and IL-13 stimulation, secrete insulin-like growth factor (IGF), fibronectin, and TGF-β, with a strong modulatory effect on NK cell function. Anti-inflammatory activity has also been ascribed to IL-10 highly secreted by M2c macrophages obtained upon stimulation with IL-10 itself, glucocorticoids, or TGF-β. However, due to the complexity of the tissue microenvironment and cell plasticity, the in vivo macrophages do not exactly mirror the simplistic in vitro scenario. In vivo tumor-associated macrophages (TAMs) can express markers belonging to both M1 or M2 cells, and can change their phenotypic and functional properties increasing their heterogeneity over time due to microenvironment perturbations related to infections or therapeutic agents [17,18]. Using last-generation techniques, many studies focused on deciphering macrophage heterogeneity in vivo. Single-cell analysis in breast cancer highlighted seven different PD-L1pos cell clusters, whose relative abundance is different in Estrogen Receptor (ER)-positive or -negative tumors, and in tumors with different grades [19]. It becomes relevant to correlate different macrophage phenotypes/signatures with different stages of the disease, and resistance or response to therapies [15]. This is crucial to plan a specific targeting of the macrophage populations more significantly associated with worse prognosis. Along this line, the heterogenous TAMs might rely on their origin. TAMs can derive from monocytes recruited in the TME and others from fetal yolk sac precursors [20]. A recent study in murine pancreatic ductal adenocarcinoma (PDAC) models showed that TAMs derived from fetal precursors have a pro-tumoral function whereas their bone marrow (BM) counterpart mainly contributes to antigen presentation [21]. In humans and mice, markers useful to distinguish between these two cell types have been described, including CX3CR1, CXCR4, CD49d, CD11b, and CD11a [21,22], and others are under investigation to enrich the signatures. Along this line, most macrophages present in the peritoneal fluid of patients with ovarian cancer express a membrane-associated form of IL-18 (mIL-18) [23]. This subset was initially detected in vitro in monocytes-derived M0 and M2 macrophages from healthy donors [24], regardless of the presence or absence of CD16 on monocytes. Due to possible differentiation from circulating monocytes, the spectrum of macrophage polarization has been increasingly studied in relationship with monocyte heterogeneity, which has been further strengthened by recent studies [25,26]. Macrophage heterogeneity should also be considered when attempting to potentiate both macrophage- and NK cell-mediated anti-tumor activity. Functional crosstalk between human macrophage and NK cells occurs [23,27,28,29,30], which should be revised based on the heterogeneity of both innate effectors. Looking at the NK cell story, these lymphocytes have rapidly moved from being “null cells” to crucial innate effectors characterized by a huge and variegated repertoire of inhibitory, activating, chemokine, and residency receptors, as well as by the existence of different circulating or tissue-resident NK cell subsets [31,32,33,34,35].

Despite the growing information about macrophages and NK cells colonizing adult TME, relatively little information is available on the “Immunoscore” of pediatric tumors such as neuroblastoma (NB). NB represents rare cancer with a very dismal prognosis, particularly in High Risk (HR) cases identified as children with metastatic disease and age > 18 months, or with localized MYC-N amplified tumors. These patients receive an aggressive combined standard therapy having a maintenance phase based on the infusion of a chimeric monoclonal antibody (mAb) targeting the GD2 antigen [36,37]. Immunotherapy aims at controlling the disease residual from the more conventional therapies (surgery, chemo, and radiotherapy) by killing tumor cells through complement activation and triggering the cytotoxicity of FcγRpos cells, such as macrophages and NK cells. Unfortunately, however, the anti-GD2 immunotherapy does not cause long-lasting benefits and more than 50% of HR-NB patients fatally relapse within five years. Whether some properties of phagocytes and NK cells have been positively correlated with a better response, including the presence at genotypic levels of FcγRs polymorphism with different affinities [36,38], other “immune responsiveness signatures” are far from being identified. These could include the presence of specific macrophage and NK cell populations present at the tumor site.

With this scenario, the review aims to sum up and discuss the most relevant data on macrophages in NB, looking at future promising immunotherapeutic strategies able to potentiate their antitumor activity and their crosstalk with other cell types colonizing TME, particularly NK cells.

## 2. Macrophages in Neuroblastoma Microenvironment

Most pre-clinical immunotherapeutic approaches against NB have been tested administrating to mice syngeneic tumor cell lines. More recently, the Tyrosine hydroxylase-MYCN (TH-MYCN) transgenic mouse model has been adopted. Although lacking spontaneous metastasis, TH-MYCN transgenic mice overexpress MYCN under the control of the tyrosine hydroxylase promoter, presenting aggressive tumors that recapitulate the location, histology, biology, and cytogenetics abnormalities of human NBs.

An immortalized 9464D cell line was derived from a spontaneous NB arising in TH-MYCN transgenic C57BL/6 mice. It grew much more quickly when injected intra-adrenally (IA) in TH-MYCN mice as compared to the subcutaneous (SC) injection. Moreover, intra-adrenal tumors were much more densely infiltrated by TAMs, which expressed low levels of MHC class II and displayed a more immunosuppressive M2-like phenotype [39]. These models were also utilized to test immunotherapies. After treatment with cyclophosphamide to create a therapeutic window of minimum residual disease allowing host immune development, it was observed that immune cell infiltration was dramatically different between IA and SC murine NB models. While showing similar GD2 and MHC class I expression, IA tumors showed a type of immune infiltration more similar to that observed in human cancers. Cyclophosphamide was also administered to TH-MYCN transgenic in combination with an anti-GD2 or anti-4-1BB monoclonal antibody (mAb). In both combination regiments, increased survival was observed. Thus, data indicate that the TH-MYCN transgenic mouse represents a suitable model for investigating NB immunobiology and testing immunotherapies in a preclinical scenario [40].

Recently, patient-derived orthotopic xenografts (PDXs) have been proposed as preclinical models more reliable than cell line-derived xenografts since they would better predict clinical outcomes. Undissociated tumor fragments from HR-NB patients were implanted into the para-adrenal area of immunocompromised NOD/SCID/gamma(c)(null) (NSG) mice. PDXs reproduced the genetic and histological features of original tumors cells and were capable of metastasizing to lungs, liver, and BM. The main TME hallmarks of the aggressive parental tumors, such as the presence of abundant cancer-associated fibroblasts (CAFs), TAMs, extracellular matrix (ECM) components, pericyte lining, and abundance of lymphatic and blood vessel vascularization, were maintained. However, information is still lacking about the survival capability of co-engrafted human tumor stroma, and the relative contribution of the human and murine stroma. In this context, in PDXs from both MYCN-amplified and -non-amplified tumors, Braekeveldt et. al. observed infiltration of mouse F4/80pos macrophages but not positivity for the human macrophage CD68 marker. This suggests the involvement of murine stroma in the tumor formation and lack of survival of the human stromal counterpart [41]. Given the relevance of the human TME in the disease progression, NB-PDXs need to be optimized to more precisely predict the efficacy of current and novel anti-cancer immunotherapies.

It should not be disregarded that some cancers can develop after a period of protracted chronic inflammation caused by microbial infections or non-biological events, such as physical or chemical stress; this would occur more in adults than in very young children, such as those affected by NB. This could have an impact on the TME composition, which seems to be quite different in adult and pediatric cancers in terms of infiltrating inflammatory cells. While in adults tumor-infiltrating cells typically include a variety of leukocytes, in pediatric ones the majority of cells are represented by macrophages, which tend to accumulate in necrotic regions [42,43]. Moreover, in the NB microenvironment, differences in cell composition and functions could be also related to the heterogenic nature of the tumor, which arise from errors in the neural crests’ differentiation program during the early phase of embryonic development. It is conceivable that tumor growth and immune cell development could occur simultaneously [44]. It is known that NBs have an immunosuppressive TME, related at least in part to MYCN amplification [45]. This allows cancer cells to evade host immune responses. In a cohort of 102 non-MYCN-amplified, untreated, primary NB tumors, high levels of inflammation-related genes characterizing M2 macrophages and a restricted gene signature (IL-6, IL-6R, IL-10, and TGF-β) were found to correlate with a worse prognosis [46]. In HR-NB patients, cancer promoting macrophages predominate both in locoregional tumors, showing high expression of CD163 and CD206 M2 markers [42], and in metastatic NBs characterized by a high presence of TAMs. Moreover, a TAM-associated gene signature, including CD33/CD16/IL-6R/IL-10/FCγR3 genes, was more frequently detected in metastatic patients lacking MYCN amplification and diagnosed at age ≥ 18 months than in patients diagnosed at age <18 months. The signature above contributed to 25% of the accuracy of a novel 14-gene-based tumor classification score significantly correlated with a worse five-year progress free survival [47].

Crucial questions remain. How TAMs can contribute to the development of NB and what are the main mechanisms acting at the level of the TME? In this context, it has been observed in several types of malignancies, including HR-NB, that high levels of IL-6 and the soluble form of its receptor (sIL-6R) in the patient’s blood and BM, correlated with bad prognosis [48,49] and would support tumor growth [50]. Furthermore, monocyte-derived IL-6 and sIL-6R activate STAT3 in NB cell lines promoting drug resistance. NB cell lines pretreated with IL-6 showed a remarkable increase in survival rate when exposed to chemotherapeutic agents. The effect was boosted by the addition of human monocyte-derived IL-6R known to have a trans-acting agonistic effect [51] and was linked to the upregulation of survival factors, such as survivin (BIRC5) and Bcl-xL (BCL2L1). Accordingly, the protection from drug-induced apoptosis was lost in the presence of STAT3 inhibitors or STAT3 gene knockdown. These data provide new insights into the role of monocytes in promoting resistance of NB to cytotoxic effects of therapeutic agents through STAT3 activation [52]. This opens the way to the possible targeting of TME inflammation-associated biologic pathways in NB. In this context, anti-IL-6 mAb have been already tested in adult cancers [53].

It has also been investigated whether IL-6 released by TAMs influenced the proliferation of NB cells through STAT3 activation and up-regulation of the c-MYC transcription. Surprisingly, slow down of NB growth, reduction of STAT3 activation, and c-MYC up-regulation were not observed in vitro, blocking IL-6, or in IL-6 knockout mice. On the contrary blocking of JAK-STAT activation, greatly inhibited the TAMs-sustained development of NBs implanted subcutaneously in NSG mice. c-MYC protein levels were also partially reduced by the inhibition of STAT3 phosphorylation, indicating that TAMs can affect NB proliferation stimulating c-MYC expression by a STAT3- and IL-6-independent mechanism [54].

All the findings above highlight the relevance of the functional interactions between tumor cells and TAMs, which should be further explored and possibly targeted in NB treatment. NBs are constitutively characterized by low expression of MHC class I [55,56,57,58], and easily evade the killing activity of cytotoxic T cells. Therefore, potentiating T-independent anti-tumor responses could represent a more effective approach limiting NB growth. Along this line, Buhtoiarov et al. analyzed mice engrafted with the NXS2 mouse NB cell line and treated with the anti-CD40 mAb. Macrophages increased the expression of intracellular toll-like receptor 9 (TLR9), becoming more sensitive to CpG-containing oligodeoxynucleotides (CpG), a TLR9 agonist. This effect was accompanied by an increased release of IFN-γ, IL-12, and TNF-α by phagocytes, and significant inhibition of tumor growth [59]. Moreover, in NSG or NOD/SCID immunodeficient mice, the depletion of monocytes/macrophages through blockade of colony stimulating factor 1 receptor (CSF1R, also known as macrophage colony-stimulating factor receptor, M-CSFR) significantly enhanced the cyclophosphamide plus topotecan combination therapy. This is in line with the in vitro observation that, topotecan can increase the release of CSF-1 (M-CSF) by NB cells, favoring TAMs differentiation [60]. Previous studies also suggested that CFS-1R blockade could antagonize the activity of CSF-1 released by stromal cells in response to chemotherapy [61,62]. Overall, these studies point out the central role of TAMs in favoring NB growth and resistance to pharmaceutical treatments.

It is of note that the TME consists of a complex network of tumor and different non-malignant cells, all of which can have fundamental interplays with macrophages. In this context, studies by Hashimoto and colleagues provided important data supporting the cooperation of TAMs and CAFs in supporting tumor progression. They demonstrated that there is a reciprocal influence among NB cells, TAM-like macrophages (CD68pos, CD163pos, CD204pos), and CAFs, identified as alpha smooth muscle actin (αSMA) positive cells. In in vitro experiments, PBMC-derived macrophages and BM-derived mesenchymal stem cells differentiated into TAM-like and CAF-like cells, respectively, after being attracted by the NB cell line. In turn, TAMs and CAFs colonization increased tumor invasiveness and growth. Moreover, TAM-like macrophages significantly promote CAFs proliferation, resulting in a synergistic effect favoring NB progression [63]. Thus, TAMs and CAFs may serve as prognostic indicators and possible therapeutic targets in NBs. Their abundance together with the co-presence of mesenchymal stromal cells (MSCs), correlated in human NBs with high histological malignancy and low T and NK cell infiltration [63,64]. Still, how NB, MSCs, CAFs, and monocytes/macrophages collaborate in establishing a pro-tumorigenic TME and supporting immune evasion is poorly understood. In a recent study, the interaction of monocytes and MSCs with NB cells was shown to cause significant and peculiar upregulation of several pro-tumorigenic factors including TGF-β1 and IL-6, which protects monocytes from spontaneous apoptosis, promoting TAMs differentiation. This cross-talk has been confirmed in both xenotransplanted tumors and primary tumors from patients. It was also shown that there was a strong correlation between the presence of CAFs and the activation in NB of p-SMAD2 and p-STAT3, which participate in the TGF-β1 and IL-6 signal, respectively [65].

With these premises, new strategies targeting the stromal compartment may heavily impact the clinical outcomes of HR-NB patients. This possibility drive studies aimed at better clarifying the cellular properties of TME including metabolic pathways regulating their activity. In this regard, arginase 2 (ARG2), which regulates arginine metabolism, was found to play a key role in NB proliferation and the establishment of immunosuppressive TME. In particular, a reciprocal crosstalk between cancer and immune cells occurred regulating ARG2 expression in NB cells and favoring tumor development. In contrast with most studies, Fultang and colleagues showed that NB-conditioned PBMC-derived monocytes turned into CD206neg Arg1low cells, a phenotype that the authors defined as M1-like; cells produced IL-1β and TNF-α that, in turn, stimulated ARG2 expression in NB. Accordingly, in patients with stage IV NB, the presence of an IL-1β and TNF-α enriched TME correlated with a worse prognosis. These findings suggest a clinically exploitable, immunological metabolic regulatory loop between tumor cells and myeloid cells regulating ARG2 [66]. The unusual detection of M1-like macrophages in NB specimens might further underline the complexity and heterogeneity of the human NB microenvironment and suggest that different areas of the tumors could be colonized by cells with different functional properties.

Regarding lipidic metabolism, the fatty acid binding protein 4 (FABP4) expression in TAMs was associated with advanced clinical stages and adverse NB histology. Invasion, migration, and growth of NB were all accelerated by FABP4pos macrophages. In macrophages, FABP4 physically interacted with ATP Synthase B (ATPB), leading to its ubiquitination with the reduction of ATP levels, deactivation of the NF-κB/RelA-IL-1α pathway, and reprogramming towards an anti-inflammatory phenotype. Thus, FABP4 could be considered a new functional marker of protumor TAMs in NB, and a possible target in immunotherapeutic approaches [67].

Other strategies to limit the pro-tumoral macrophage activity could consist in hampering monocyte recruitment at tumor sites. In this context, the lipid sphingosine-1-phosphate (S1P) was shown to promote in NB the expression of the CCL2 chemokine, known to attract monocytes to inflammation sites. Blocking the S1P2 downstream signal by selective antagonists reduced CCL2 expression, and resulted in a remarkable reduction of F4/80pos macrophages in NB xenograft, and decreased tumor growth [68,69]. In humans, non-cytotoxic doses of the tyrosine kinase inhibitors (TKI) imatinib and nilotinib caused interesting off-target effects, including the reduction in the expression in monocytes of CCR1 and CSF-1, and inhibition of their differentiation towards macrophages [70].

Efforts have been made to understand how to effectively reprogram the M2-like macrophages towards an M1-like, anti-tumoral, phenotype. Bacillus Calmette-Guerin (BCG) can induce M1 polarization of M0 macrophages and revert that of M2 [27], and BCG treatment is increasingly accepted by multiple guidelines for invasive bladder cancer [71].

A recent work published by Relation et al. engineered MSCs to produce and release IFN-γ at the tumor site. This strategy led to the transient polarization of macrophages toward the M1-like phenotype (expressing IL-17 and IL-23p19) in orthotopic NB xenografts, with reduced tumor growth and increased overall survival, without any systemic toxicity [72]. CAF-derived prostaglandin E2 (PGE2) stimulates NB growth and alters immune responses via a variety of mechanisms [73,74], including the induction of M2 macrophage polarization. The inhibition of PGE2 in TH-MYCN transgenic mice reprogrammed macrophages to M1 phenotype and reduced NB growth, angiogenesis, and CAFs infiltration [75].

Interactions between different cellular types occurring in the NB TME have been represented in Figure 1, while innovative therapeutic approaches in the pre-clinical scenario targeting the monocyte/macrophage compartment have been summarized in Table 1.

## 3. Macrophages and Natural Killer Cells Crosstalk

As discussed above, strategies aimed at reprogramming TAMs represent a promising approach. This is also due to the positive effect of M1 polarized macrophages on NK cell functions, as demonstrated by studies in mouse models and humans. In particular, TLR agonists, such as LPS or BCG, engage M2 and TAMs, inducing their polarization toward M1 that, in turn, activate human NK cells, as demonstrated by the increased cytotoxic function and IFN-γ release [23,27]. Most of the effects require NK-to-macrophages contacts. In this context, the interactions between DNAM-1 and 2B4, on NK, and their ligands on macrophages play a fundamental role. sIL-18 released during M1 polarization provides a significant contribution to NK cell activation that was compromised by mAbs blocking either the cytokine or the specific receptor. Interestingly, treatment with monensin, which hampers intracellular protein transport, indicates that the sIL-18 released could derive by shedding of the membrane form. mIL-18, expressed on the cell surface of M0, M2, and TAM [23,24], is lost upon TLR activation, a phenomenon that is paralleled by the detection of the soluble form in the supernatant and NK cell activation. The mIL-18 expression is induced by M-CSF in a subpopulation (30–40%) of macrophages differentiating from both CD16neg and CD16pos monocytes; while it is undetectable in monocytes, GM-CSF-treated monocytes, and monocyte-derived DC. mIL-18 expression is significantly reduced by the treatment with the caspase-1 inhibitor suggesting the requirement of an assembled inflammasome for IL-18 surface expression (Figure 2). Interestingly, high percentages (up to 90%) of macrophages present in the peritoneal fluid of ovarian cancer patients expressed mIL-18, suggesting a possible role in TME [23].

In addition to TAMs reprogramming, the activation of NK cells may reduce in vivo the number of TAMs. In vitro experiments showed that properly activated NK cells, isolated from PBMC and peritoneal fluid of ovarian cancer patients, efficiently killed autologous TAMs, which were characterized by low, “non-protective” levels of MHC class I molecules [23]. Mattiola I. and collaborators described another macrophage/NK functional interaction potentially relevant for future therapeutic approaches [28]. This involves the membrane-spanning four domains A4A (MS4A4A) molecule whose expression is detected in CD163pos TAMs. MS4A4A colocalizes with dectin-1 in lipid rafts and is crucial to support optimal Syk phosphorylation and dectin-1 functions such as the production of inflammatory cytokines and reactive oxygen species. Importantly, dectin-1 induces on macrophages the expression of IFN regulatory factor 3-dependent NK-activating molecule (INAM, also known as Fam26), promoting NK cell activation with increased tumor cell killing.

Other strategies potentiating the NK cell function could involve methods blocking the immunosuppressive loops occurring in TME, particularly those involving the monocyte and macrophages compartment. As already discussed above, IL-6, released by mononuclear phagocytes upon NB conditioning, and TGF-β1, produced by both immune and tumor cells, inhibit the IL-2-mediated activation of NK cells, through the activation of the STAT3 and SMAD2/3 pathways and suppression of IFN-γ, granzymes, and perforin release. This is in line with the well-documented regulatory role of TGF-β in NK cell activation that emerged from several studies [32,76] supporting the development in NB of preclinical [77] and clinical studies with combining immunotherapies including the block of the TGF-β activity. Importantly, in NK cells, TGF-β also decreased the expression of activating receptors involved in NB recognition and modified the chemokine receptor repertoire, likely hampering the NK cell recruitment at the tumor sites [78]. This observation could have in vivo a profound pathophysiological impact.

The use of immune-modulating drugs was also revealed to be effective in restoring NK cell activity. Lenalidomide, which is known to induce in T cells the secretion of IL-2, IFN-γ, and TNF-α, showed promising results in several pre-clinical cancer models in combination with mAbs inducing in NK cells antibody-dependent cytotoxicity (ADCC) (e.g., anti-CD20 in lymphoma and chronic lymphocytic leukemia) [79,80,81] and in clinical trials both in adult and children, with an increased number of cytotoxic NK cells [82,83]. Moreover, in in vitro and in NOD/SCID mouse models, lenalidomide blocked the adverse effects of both IL-6 and TGF-β1, adjuvating the anti-tumor effect of anti-GD2 immunotherapy [84]. Along this line, the combination of histone deacetylase inhibitors (HDACi, Vorinostat) and anti-GD2 immunotherapy is also presently being investigated with encouraging results. In an aggressive orthotopic mouse model, the combined approach increased NB cell death and shaped tumor and stromal cell phenotype and composition. In particular, tumor cells surviving the drug treatment increased the expression of GD2, and TME was characterized by a high number of macrophages, expressing high amounts of MHC class II and FcRs, and a reduced quantity of myeloid-derived suppressor cells (MDSC). Collectively, these data provide a rationale for the clinical testing of anti-GD2 mAbs and Vorinostat combining therapy in NB patients [85,86].

Strategies boosting NK cell cytotoxicity and reducing the pro-tumoral effects of macrophages and other suppressor cells may represent promising adjuvants potentiating standard immunotherapy. Along this line, in a recently published study, the anti-GD2 mAb hu14.18 has been linked to IL-15 or IL-21 immunostimulatory cytokines. In immunocompetent mice engrafted with syngeneic NB, this approach enhanced NK cell-mediated ADCC against NB; it also increased in TME the number of CD8pos T cells and M1-polarized TAMs, while decreasing that of regulatory T cells and MDSC [87]. Current immunotherapy mad be also influenced by tumor-derived small extracellular vesicles highlighted the role of tumor-derived small extracellular vesicles (sEV). Liu et al. highlighted the role of sEVs as crucial mediators regulating responses to immunotherapy demonstrating that NB-derived sEV attenuated the in vivo effectiveness of the anti-GD2 mAb dinutuximab (Qarziba) and promoted an immunosuppressive TME rich in TAMs and poor in NK cells. NB-sEVs were also able to block anti-GD2-mediated NK cell ADCC in vitro and splenic NK cell maturation in vivo. When sEVs secretion was pharmacologically reduced using tipifarnib, an FDA-approved farnesyltransferase inhibitor, a significant improvement in the dinutuximab efficacy was observed with reduced tumor growth and immunosuppressive environment [88]. Another possibility to increase the effectiveness of the anti-GD2 immunotherapy could be contrasting the detrimental effects of MSCs. In this context, MSCs and monocytes promoted NB growth and negatively affected ADCC mediated by dinutuximab-activated NK cells, both in vitro and in NSG mice, and using NB cell lines and PDX. This detrimental effect was efficiently antagonized by anti-CD105 antibodies that depleted MSCs, endothelial cells, and macrophages from the TME [89].

## 4. Conclusions and Future Directions

HR-NB represents a worldwide emergency due to the high failure rate of those patients who do not respond to the current standard therapy. It is commonly recognized that non-malignant cells, residing or recruited within the tumor site, are fundamental for the development and growth of tumors such as NB. The TME, which includes immune cells, also supports cancer cells in evading the anti-tumor activity of the immune system. As discussed above, a crucial role is played by the myeloid compartment. In particular, macrophages assume an anti-inflammatory M2-like phenotype, which promotes tumor progression through a reciprocal crosstalk. This often correlates with a worse prognosis in HR-NB patients. Several studies investigated a variety of approaches possibly contrasting these tumor-promoting effects in different cancers including NB; they comprise the reprogramming of macrophage polarization toward M1, enhancement of mAb-dependent phagocytosis, and reinforcement of the NK-mediated cytotoxicity by the standard clinically approved anti-GD2 mAbs used alone or in the combination with other therapeutics [90].

The understanding of the heterogeneity of TAMs within the TME still remains a challenge. It is widely accepted that the original M1/M2 dichotomy in macrophage polarization is an oversimplification and new techniques will contribute to the puzzle solving. For example, the proteomic analysis could identify molecules differentially expressed in the various macrophage subsets; two of these molecules, mIL18 and MS4A4A, have been identified, whose role in macrophage heterogeneity needs to be clarified. It will be relevant to understand the prognostic value of the macrophage subpopulations and correlate their phenotypic/functional properties with anti-tumor responses and, in particular, the capability of modulating NK cell activity. In this regard, it will be also crucial to better investigate the phenotypic and functional heterogeneity of NK cells infiltrating NB to design novel and more effective therapeutic approaches simultaneously enhancing the anti-tumoral activity of NK cells and macrophages as well as their reciprocal crosstalk. Decisive is adding information on the NK [58,91,92] and macrophage landscape in tissues, particularly in the BM, the most frequent site of NB metastasis and relapses. The strategies aimed at potentiating macrophage/NK interactions should also consider the possible modulation of molecules negatively regulating their function. In this context, the currently exploited therapeutic approach can result in undesired side effects, such as the upregulation of immune checkpoints [93,94,95,96], observations that are guiding the choice of promising combination therapeutic approaches.

Many efforts are also dedicated to find alternative effective therapeutic approaches with less toxicity. For example, O-acetylated GD2 (OAcGD2) is a promising novel tumor-associated molecule that is not expressed by peripheral nerves, being targetable with reduced painful side effects. In a pre-clinical setting, the anti-OAcGD2 mAb activated the immune system and increased the macrophage infiltration/function within the TME. However, the treatment efficacy was hampered by the upregulation on NB cells of CD47 [95], which interacts with the SIRPα receptor on macrophages, limiting phagocytosis [97,98,99]. This further highlights the need to increasingly evaluate the use of combined therapies.

Various mechanisms occur within the TME favoring cancer progression, which are often mediated by stromal and immune cells. In this context, even if the T cell-mediated surveillance could be poorly relevant in NB lacking MHC-I expression, T cells represent essential effectors in cancer immunotherapy and are subjected to modulation by specific pathways arising within the TME. For example, human cancers characterized by a poor T cell infiltration showed strong activation of the WNT/β-catenin pathway [100,101,102,103,104]. This pathway is involved in T cell exclusion, as well as in tumor progression, invasion, and metastasis. In mouse models of ovarian cancer, the inhibition of this pathway decreased tumor progression, enhanced the survival, and increased the number of CD8pos T cells within the TME [105,106]. In NB, WNT/β-catenin plays a pivotal role in cellular proliferation and apoptosis as well as in the embryonic development, with implications in NB onset, progression, and relapse. Its activation also enhances MYCN amplification and favors chemoresistance [107,108]. Wang et al. demonstrated that MYCN knockdown in NB cell lines remarkably reduced cell viability, accelerated apoptosis, and blocked WNT/β-catenin signaling [109]. Several factors seem to interact with this pathway inducing in NB cells a malignant phenotype, cancer stemness, or epithelial-to-mesenchymal transition (EMT). These include the nucleotide binding oncotarget BORIS [110], cell surface proteoglycan Glypican-2 (GPC2) [111], and transmembrane protein human tripartite motif 59 (TRIM59) [112]. WNT/β-catenin was shown also to regulate the CAFs activity [113] and macrophage interactions with tumors. In hepatic tumors, reciprocal influence between cancer cells and phagocytes has been demonstrated; TNF-α produced by TAMs induced EMT and stemness in liver tumor cells [114], the latter in turn promoted M2 macrophage polarization via the Wnt/β-catenin signaling [115]. It was also reported that macrophage-derived soluble factors activated the WNT signaling pathway in colorectal cancer [116]. For instance, tumor cells stimulated macrophages to release IL-1β, which enhanced the levels of β-catenin, resulting in higher expression of WNT target genes in cancer cells [102,103,116]. Furthermore, macrophage-induced IL-6 favored the migration and invasion of colon cancer cell via WNT/β-catenin in a STAT3/ERK-dependent manner [117]. As recently investigated in mice, WNT/β-catenin signaling blockade might be used in combination with therapeutic strategies, limiting the expression of inhibitory ligands in cancer cells such as CD47 [101] or blocking PD-1/PD-Ls interactions [118].

Novel therapeutic approaches need to be tested in highly predictive preclinical platforms, a crucial step when investigating new drugs. However, the currently available tools are only partially reliable and present limitations that could explain the high rate of failure in translating novel approaches into clinics. As mentioned above, most studies have been conducted using human NB cell lines or various mouse models. Currently used long-term NB cell lines have been extensively characterized. However, they could develop genetic alterations and undergo clonal selection due to their prolonged expansion in 2D cultures, acquiring phenotypic and functional properties far from the original tumor. The in vivo models have been largely used in cancer research. Different kinds of mouse models have also been exploited to understand the role of the TME in NB immune resistance, with a particular focus on TAMs. Syngeneic, orthotopic, and transgenic NB models, as well as PDXs, have been largely used to study disease progression and validate innovative therapeutic approaches. However, researchers are aware that even the more complex mouse models have important limitations: the great variety of incidence of metastases among different models, which is often dramatically poor; the absence in NOD-SCID and NSG mouse of immune pressure due to the lack of T, B, and NK cells; the presence in commonly used immunodeficient mice models of an incomplete TME containing cells (macrophages, fibroblasts) of mouse origin. In this context, it is unlikely that PDXs could maintain their original characteristics and, surely, could not follow the tumor evolution occurring in the patient.

The host xenogeneic stroma replacing the human counterpart in implanted tumor specimens hampers the possibility to analyze the interactions occurring among NB and stromal and immune cells. Therefore, the scientific community is working hard to realize more complex and reliable tools. These include three-dimensional (3D) cell cultures or organ-on-chip in vitro models allowing cells to grow in a spatial organization more similar to in vivo tissues and to experience dynamic stimuli, e.g., mechanical stimulations, and fluid flow, occurring in different human organs. Relevant 3D systems have been developed in recent years and promising results have been obtained that will be useful to better address the complex functional crosstalk occurring at the tumor site [94,119,120,121,122,123,124,125,126]. In addition, the development of “humanized” mice generated by the transplantation of human hematopoietic cells into immune-compromised mice could help in studying the in vivo human immune response in tumors and during inflammation. Indeed, the so-called MISTRG mice robustly develop multiple immune cell types, including macrophages, neutrophils, dendritic cells (DC), and NK cells [127,128].

Although important steps forward in the discovery of novel cures have been made, HR-NB still represents a challenge. Given the high heterogeneity of this tumor, strategies may lead to good responses in specific subgroups of patients while being ineffective in others [129]. Therefore, to personalize the diagnostic and therapeutic approach, it is mandatory to better characterize the TME, taking into consideration the tumor, stroma, and immune compartment as well as their molecular and functional crosstalk.

## Figures and Tables

**Figure 1 cells-12-00885-f001:**
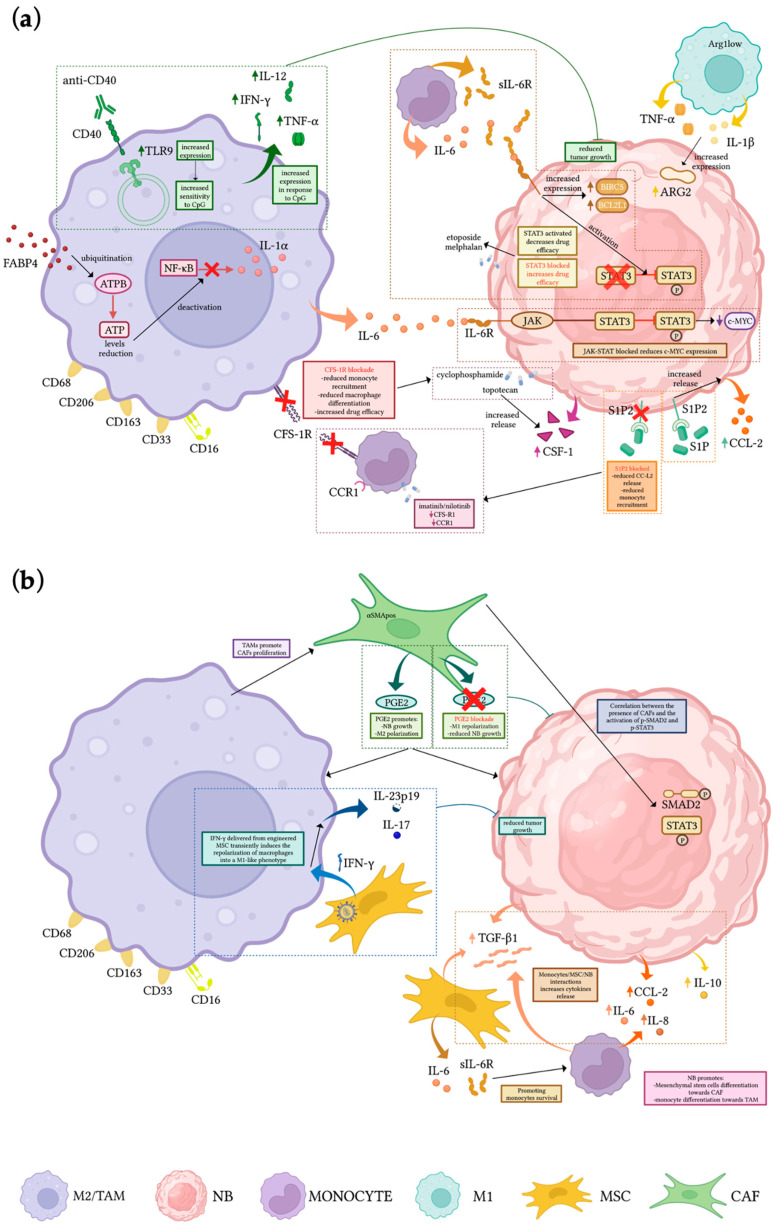
Cellular interactions in the NB microenvironment (**a**,**b**). Mechanisms of crosstalk among NB cells and macrophages/TAMs, monocytes, and stromal cells, and therapeutic strategies aimed at reducing tumor growth, reprogramming macrophage polarization, or reducing monocytes recruitment at the tumor site. Created with BioRender.com (accessed on 12 February 2023).

**Figure 2 cells-12-00885-f002:**
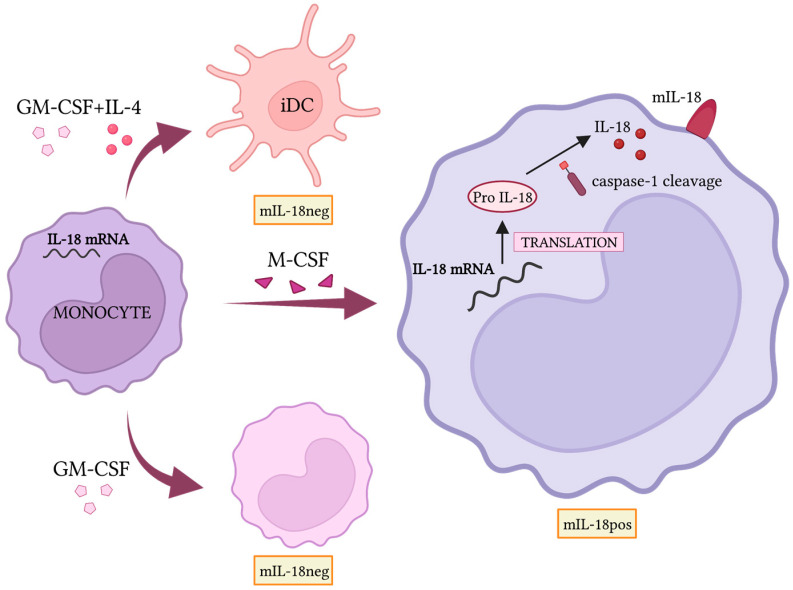
Mechanisms allowing mIL-18 expression. Monocytes from peripheral blood of healthy donors don’t express mIL-18, although IL-18 mRNA and pro-IL-18 is detected. Monocyte treatment with GM-CSF or their differentiation toward DCs (GM-CSF + IL-4) doesn’t result in mIL-18 expression. On the contrary, the M-CSF-induced monocyte differentiation toward macrophages is accompanied by mIL-18 expression in a subpopulation (30–40%) of unpolarized macrophages Created with BioRender.com (accessed on 12 February 2023).

**Table 1 cells-12-00885-t001:** Therapeutic strategy targeting monocyte/macrophages in NB TME in pre-clinical settings.

Monocyte/Macrophages-Related Pathway/Role in NB TME	Pre-Clinical Model	Treatment	Outcomes	Ref
Monocyte-derived IL-6 and sIL-6R activate STAT3 in NB cells enhancing drug resistance, increasing survival, and upregulating NIRC5 and BCL2L1	In vitro cell culture	STAT3 inhibitors/gene knockdown	Activation of STAT3 in NB cells enhancing drug resistance; increased survival rate; upregulation of BIRC5 and BCL2L1	[52]
TAMs affect NB proliferation stimulating c-MYC expression	NSG mice	Block of JAK-STAT activation	Inhibition of TAMs-induced NB growth and partial reduction of c-MYC	[54]
Cytotoxic activity of unpolarized macrophages can be activated	In vitro cell culture; NSX2 NOD/SCID xenograft mouse	Anti-CD40 mAbs and CpG	Increased expression of TLR9 in macrophages becoming more sensitive to CpG and increasing the release of IFN-γ, IL-12, and TNF-α, resulting in the inhibition of tumor growth	[59]
TAMs differentiation is favored by CSF-1 released by NB cells upon topotecan treatment	NOD/SCID xenograft mouse	Blockade of CSF1R	Enhanced effect of chemotherapy, resulting in increased survival and decreased tumor growth	[60]
CXCL2 released from TAMs increases NB invasiveness via CXCL2/CXCR2 axis	In vitro cell culture	Neutralization of CXCR2 in NB cells	Decrease effect of TAMs on the invasiveness of NB cells	[63]
Interaction of monocytes and MSCs with NB leads to the upregulation of TGF-β1 and IL-6, protecting monocytes from apoptosis and promoting TAMs differentiation	In vitro cell culture; NB xenograft	Blockade of IL-6R (Tocilizumab) and TGF-βR1 (Galunisertib)	Loss of anti-apoptotic effect of MSC on MN is suppressed by blocking IL-6R but not TGF-βR1	[65]
M1-like macrophages (CD206neg Arg1low) release IL-1β and TNF-α inducing ARG2 expression in NB	In vitro cell culture/TH-MYCN transgenic mouse	Blockade of arginine metabolism	Decrease NB proliferation in vitro, delayed progression and increased survival in vivo	[66]
FABP4pos macrophages promote invasiveness and growth of NB and are associated with poor prognosis	In vitro cell culture; NB tissue sample	FABP4-knockdown	Promotion of CD80posCCR7pos phenotype and tumor-inhibiting effect through NF-κB/RelA-mediated IL-1α regulation.	[67]
CCL2 released by NB cells attract monocytes within the tumor site	NB xenograft	Blockade of S1P2 downstream signal of S1P	Reduced CCL2 expression in NB resulting in reduction of F4/80pos macrophages and decreased tumor growth	[68,69]
TAMs can be repolarized into an M1-like phenotype	In vitro cell culture; NB xenograft	Engineering MSCs to produce and locally release IFN-γ	Transient polarization of macrophages towards M1-like phenotype (expression of IL-17 and IL-23p19, reduced tumor growth and increased survival	[72]
CAFs-derived PGE2 promotes M2 macrophage polarization	TH-MYCN transgenic mice	Inhibition of PGE2	Reprogramming of macrophages to M1 phenotype, reduced NB growth, angiogenesis and CAFs infiltration	[75]

## Data Availability

Not applicable.

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
