# Peer review of "Monocyte and Macrophage in Neuroblastoma: Blocking Their Pro-Tumoral Functions and Strengthening Their Crosstalk with Natural Killer Cells"

_cells, 2023, doi:10.3390/cells12060885_

Round 1

Reviewer 1 Report

In the present review entitled „Monocyte and Macrophage in Neuroblastoma: Blocking Their Pro-Tumoral Functions and Strengthening Their Crosstalk with Natural Killer Cells,” the Authors well describe the immunological problems that appear in HR-NB and the difficulties of the treatments.

The manuscript is well-written, it is easy to follow and summarizes the cellular interactions between macrophages and NK cells as well as the modifying action of the tumour microenvironment. The Authors give details about the biological treatments and future prospects. The Authors use a remarkable number of references many from the last few years.

However, the figures are well-detailed and informative it is difficult to read them according to the font size.

Since many markers, gene signatures etc. are listed in the text, utilization of tables showing these proteins would be beneficial parallel with the figures (e.g. description of the different types of macrophages and TAMs).

It would be preferable to show the treatments (old and new and possible ones), their targets and the cellular response in a table.

Reviewer 2 Report

The authors reviewed the main mechanisms regulating the crosstalk among NB cells and TAMs or other cellular components of the TME, which support tumor development and induce drug resistance. The manuscript is nicely presented and provides a great understanding of the process. Every aspect of factors associated with TME has been taken care of. I recommend the manuscript for publication with some minor corrections.

1.      Line 46-47:  Correct the typo in “and Natural Killer Cells (NK) showing” and change to “and Natural Killer (NK) cells showing”. I would suggest checking thoroughly for such typos.

2.      Authors should add the role of the WNT/β-Catenin pathway associated with the infiltration of the inflammatory cells and other components in the tumor microenvironment. Moreover, there are reports available that demonstrate the how WNT/β-Catenin pathway can regulate the TME and inhibit the effect of checkpoint blockers. I would suggest incorporating the points and thoughts on how this pathway can be important in the management of TME.

Reviewer 3 Report

Vitale et al have produced a nice informative review on the contributions of monocytes and macrophages in neuroblastoma with a focus on cancer immunotherapy. This review will be generally found to be useful for the NB field and perhaps the larger community interested in cancer biology. The authors present adequately detailed information on the microenvironment and specifically TAMs in the context of NB. The molecular interactions between NB tumors and TAMs is described. Therapeutic strategies targeting macrophages in the tumor microenvironment are reviewed and potential new therapeutic strategies are suggested. 

The manuscript is well-written and the one figure supports and enhances the content.
